# Primary Corrosion Cause of P110S Steel Tubing Corrosion Thinning in CO$_2$–H$_2$S Well and Its Remaining Life Prediction

**Wu Long [1], Xi Wang [2,*], Huan Hu [2], Wei Lu [2], Lian Liu [1], Miaopeng Zhou [2], Sirui Cao [2] and Xiaowen Chen [2,*]**

[1] Petroleum Engineering Technology Research Institute, Sinopec Northwest Oilfield Company, Urumqi 830011, China
[2] State Key Laboratory of Oil & Gas Reservoir Geology and Exploitation, Southwest Petroleum University, Chengdu 610500, China
* Correspondence: wxxswpu@163.com (X.W.); xwchen5188@163.com (X.C.)

**Abstract:** To investigate the tube thinning of gas wells in the northwestern oilfield, the failed tubing was analyzed by using material property testing, SEM, EDS, and XRD. A novel model that was specific to the service life of tubing in terms of the wall thickness of tubing was established. The model is based on the circumferential stress of tubing. The safety factor against internal pressure and corrosion rate are considered in the model. Our results make clear that the chemical composition, non-metallic inclusion, hardness, and tensile strength of the pipe meet the requirements of relevant standards. The corrosion products on the inner and outer wall of the tubing are mainly FeCO$_3$ and BaSO$_4$, while CaCO$_3$ exists in the outer wall. Additionally, we prove that the corrosion process of the failed tubing is CO$_2$ corrosion. The tubing under the packer suffers from water-accumulation corrosion, and the tubing above the packer suffers from water-carrying corrosion. It is observed that the failed tubing arises under-deposit corrosion in local areas. According to the model calculation, the safe service life of tubing above the packer is 20.6 years. However, the safe service life of tubing below the packer is only 4.9 years.

**Keywords:** CO$_2$–H$_2$S corrosion; residual service life; under-deposit corrosion; gas well; failure analysis

## 1. Introduction

With the development of deep CO$_2$–H$_2$S reservoirs, the service environment of tubing and casing is increasingly harsh [1,2]. During the exploitation of gas fields, traces of water accumulation were found at the bottom of the hole [3]. CO$_2$–H$_2$S gas dissolves in water, causing casing and tubing at the bottom of the well to experience corrosion. Once the tubing and casing have corrosion failure, it may cause serious economic loss and environmental pollution [4].

So far, scholars have discovered the corrosion mechanism of tubing in a CO$_2$–H$_2$S environment [5–7]. Souza et al. found that FeS film decreased the CO$_2$ corrosion and delayed the precipitation of FeCO$_3$ with the existence of H$_2$S [8]. Choi et al. found that the addition of a trace of H$_2$S reduced the corrosion rate of carbon steel in the environment with high pressure CO$_2$ [9]. Javidi et al. investigated the fact that the corrosion rate increased sharply, and the pitting potential underwent a significant negative shift, when the concentration of H$_2$S increased to 200 ppm [10]. Abadeh et al. argued that the presence of 100 ppm H$_2$S concentration led to the formation of FeS as a corrosion product, which inhibited the corrosion rate [11]. Pessu et al. discovered that the pitting attack increased and became more serious at a higher level of H$_2$S content and at a higher temperature. The morphology of pitting corrosion attack is also related to the changes in the H$_2$S content [12]. From the above, scholars found that a trace of H$_2$S could slow down the CO$_2$ corrosion, while a high concentration of H$_2$S accelerated corrosion and caused pitting corrosion. Han et al. averred that there was no obvious plastic deformation on the fracture surface in the H$_2$S solution, and the fatigue cracks sprouted from the surface and expanded into the

specimen with radiation pattern [13]. However, in the actual production environment, there are still differences between the service environment of tubing and the environment of indoor experiment. In addition, many scholars were also committed to analyzing the causes of corrosion failure of oil wells with $CO_2$–$H_2S$ in oilfields. Liu et al. analyzed the stress corrosion cracking mechanism of tubing in ultra-deep wells in a $CO_2$–$H_2S$ environment [14]. Zhang et al. confirmed that the stress corrosion crack could lead to the failure of super 13 Cr tubing in high-temperature and high-pressure gas wells [15]. Ps A et al. also reported the corrosion failure of a semiconductor polycrystalline distillation column, and the main forms of corrosion for rectification towers were pitting corrosion and stress corrosion caused by chloride ions [16]. At the same time, many scholars studied the relationship between microstructure and corrosion behavior. Mousavi et al. found that the microstructure of the steel is ferritic–pearlitic together with islands of martensite/austenite constituents. The increase in heat treatment temperature led to a decrease in pearlite quantity, which ultimately led to the improvement of acid corrosion resistance of X70 and API X65 pipeline steels [17,18]. Wang et al. proved that the tensile properties of X80 pipeline steel were seriously damaged in a $H_2S/CO_2$ environment. The fracture mode changed from ductile quasi-cleavage fracture to quasi-cleavage fracture with increasing $H_2S/CO_2$ partial pressure ratio [19]. Asadian et al. found that the corrosion behavior of ASTM-A-106A pipeline steel in an acidic environment was intensified due to the increase in temperature. The EDS analysis shows that $FeCO_3$ and FeS compounds cannot form a stable protective film at high temperatures [20]. Liu et al. found that X52 Anti-$H_2S$ pipeline steel formed a double-layer corrosion film in a high-concentration $H_2S$ solution with the increase in time, which is an effective barrier to prevent diffusion. The inner layer is a fine crystal layer containing iron, and the outer layer is a columnar crystal layer containing S [21].

Moreover, the wall thickness measurement of tubing was also the key work of oilfield corrosion and protection. However, how to use the wall thickness data to predict the remaining safe life of tubing and provide guidance for the safe operation of tubing was also an urgent problem to be solved. Zhang et al. calculated the remaining safe life of tubing and casing in sour-gas wells by using corrosion rate obtained from indoor simulation experiments [22]. Dong et al. also used this model to calculate the safe service life of tubing in polymer flooding [23]. From the above analysis, the remaining safe service life of tubing was based on the corrosion rate results of indoor simulation experiments. There is still a lack of a model to calculate the safe service life of tubing and casing in the field of oil and gas wells.

Thus, we need to pay more attention to tube thinning due to $CO_2$–$H_2S$ corrosion in the actual service environment and the safe service life of tubing. Taking a gas well in the northwestern oilfield located in China as an example, the thinning reason of gas wells in a $CO_2$–$H_2S$ environment was analyzed by using the chemical composition test, metallographic test, hardness test, mechanical property test, corrosion product film observation, and 3D laser confocal test. Then, according to the test results and the service environment of the tubing, the failure reason and mechanism of tubing were inferred. Finally, a novel safety service life model of tubing based on the thickness measurement data was used to predict the residual life of the tubing. The results were very important for the safe operation of wells with $CO_2$–$H_2S$ gas.

## 2. Experiment

### 2.1. Reason Judgement of Tubing Thinning

The depth of the failed well that is located in the northwestern oilfield is 7500 m, and the well is rich in $CO_2$ and $H_2S$ gas. Among them, the content of $H_2S$ is 9318 mg/m$^3$, the partial pressure of $CO_2$ is 1.73 MPa; the temperature of bottom well is 160 °C and the water content is 0.1%. According to the thickness measurement and calculation results, the reduction in wall thickness of tubing above the packer is only 0.125 mm per year, while that of tubing below the packer is as high as 0.445 mm per year. Since the outer wall of the tubing above the packer is exposed to the annulus protection fluid, the outer wall of the

tubing is hardly corroded. Moreover, a trace of produced fluid is also found at the bottom of the wellbore. According to the location of the thinned tubing and well depth, it can be judged that the tubing under the packer is in water, leading to serious corrosion. Dong et al. pointed out that condensate, ponding in formation water, and water carried in produced gas are all types of produced water. Therefore, according to the corrosion characteristics of the well, it can be judged that the tubing above the packer is gas-carrying corrosion, while the tubing below the packer is ponding corrosion. The wellbore structure in Figure 1 shows surface casing (Φ 273.10 mm), technical casing (Φ 193.70 mm), and oil layer casing (Φ 139.70 mm) from top to bottom. There is a cement sheath between the casings Tubes 88.9 and 73 mm in diameter are used above and below the packer, respectively.

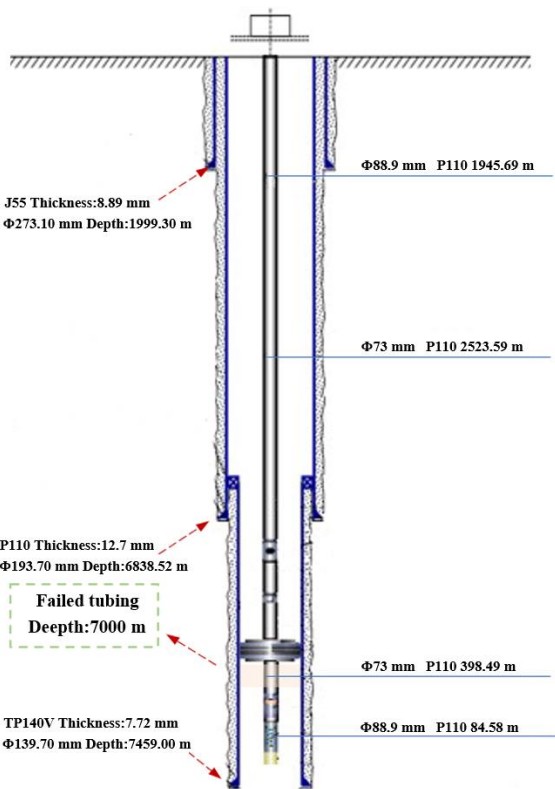

**Figure 1.** Failure position of tubing in the wellbore.

*2.2. Failed Tubing Analysis Experiment*

To reveal the reasons for tube thinning, a range of tests were performed. Among them, the chemical components test was analyzed to confirm whether the pernicious elements in the tubing exceeded the standard according to the ISO 11960-2011 standard [24]. The metallographic structure, the hardness, and the tensile strength of the tubing were tested to evaluate the quality of tubing according to ISO 11960-2011 and ASTM E45-18a standard [25]. Moreover, the surface and cross-section morphology of corrosion product film were observed by using SEM (JEOL, SEM JSM-7500F, Tokyo, Japan), the element distribution was analyzed by using EDS coupled with SEM, and the composition of corrosion products was analyzed by XRD (Cu Kα, λ = 0.154 mm, Rigaku XRD, Model D/Max-B, Tokyo, Japan) to determine the failure reason of the material. The surface morphology after removal of corrosion scales was observed by using 3D confocal laser microscope (Keyence VK-X250K, Osaka, Japan), and the roughness test area of samples was 1.25 × 1.25 mm.

*2.3. Safety Service Life Assessment of Tubing*

A calculation model for residual service life of tubing was established according to thickness measurement data of failed tubing. The thickness of the tubing became thinner after corrosion, leading to the strength property being decreased. The risk of fracture failure

of the tubing was very high when the strength safety factor was reduced to below the safety value. Therefore, it is necessary to carry out safety assessment and replace the tubing in time, so as to avoid the tubing breaking and falling off the well and affecting the safety of the well.

As the key part of failure is close to the bottom of the well, the tubing is mainly subjected to internal pressure load, and there is a relationship between the internal pressure of the tubing and the circumferential stress. Under different thinning conditions, when the circumferential stress reaches the yield strength, the corresponding internal pressure is the residual internal pressure resistance strength. When the circumferential stress exceeds the yield strength, the tubing is considered as failed.

The method of predicted residual safe service life of tubing could be described:

(1)  Based on the thickness measurement data of the failed tubing, the uniform corrosion rate of the tubing can be calculated first. According to the relationship between the internal pressure and the circumferential stress of the tubing, the residual internal pressure resistance of the tubing can be further determined.

According to API 5C3 standard [26], the circumferential stress of tubing was calculated using Equation (1).

$$\sigma = \frac{P_i R}{2\delta} \tag{1}$$

where $R$ is the external diameter of original tubing, mm; $\delta$ is the origin wall thickness of tubing, mm; $\sigma$ is the circumferential stress of tubing after service t time, MPa; and $P_i$ is the internal pressure on tubing, MPa.

Due to uniform corrosion, the wall thickness of the tubing becomes $\delta_0$ ($\delta_0 = \delta - vt$) after a period of service. At this time, $\delta$ in Equation (1) becomes $\delta - vt$, the circumferential stress of the tubing can be calculated using Equation (2).

$$\sigma = \frac{P_i R}{2[\delta - vt]} \tag{2}$$

where $t$ is the service time, year; $v$ is the uniform corrosion rate, mm/y.

If the circumferential stress ($\sigma$) reaches the yield strength of the tubing material ($\sigma_y$), the corresponding internal pressure ($P_i$) is the residual internal pressure strength ($P_{bo}$), which is shown in Equation (3).

$$P_{bo} = \frac{2\sigma_y[\delta - vt]}{R} \tag{3}$$

where $\sigma_y$ is the yield strength, MPa; $P_{bo}$ is the residual internal pressure strength of tubing, MPa.

(2)  According to API 5C3 standard, safety factor against internal pressure can be calculated by Equation (4).

$$\lambda = \frac{P_{bo}}{P_i} = \frac{2\sigma_y[\delta - vt]}{RP_i} \tag{4}$$

where $\lambda$ is safety factor against internal pressure.

(3)  According to clause 5.2.3.5.1 in AQ2012-2007 standard [27], the average threshold value of safety factor against internal pressure is 1.15, and the remaining corrosion life of tubing is shown in Equation (5).

$$T = \begin{cases} 0 & \lambda < 1.15 \\ \frac{\delta}{v} - \frac{1.15RP_i}{2\sigma_y v} & \lambda \geq 1.15 \end{cases} \tag{5}$$

where $T$ is remaining corrosion life of tubing, year.

Based on the above analysis, the model simplifies the stress load on the tubing, takes the circumferential stress of the tubing as the main criterion for failure, introduces the internal pressure safety threshold, and, finally, establishes a prediction model for the remaining life of the tubing. The model can provide guidance for the life prediction of the tubing.

## 3. Results

### 3.1. Chemical Composition

To determine whether the chemical make-up of thinned tubing material meets the requirements, the chemical composition of the failed tubing material is analyzed using direct reading spectrometer (SPECTRO MAXx (LMX06), Clive, Germany). The sample polished off the corrosion products is used to determine the element content of three areas as shown in Table 1, and the standard deviation of the element content of each parallel area is not more than 1%. The chemical composition of failed tubing meets the ISO 11960-2011 standard requirements.

**Table 1.** Chemical composition of failed tubing (wt.%).

| Chemical Composition | | C | S | Mn | P | Ni | Cr | Cu | Mo | Si |
|---|---|---|---|---|---|---|---|---|---|---|
| Content | | 0.25 | 0.003 | 0.64 | 0.006 | 0.01 | 0.96 | 0.02 | 0.95 | 0.26 |
| ISO 11960-2011 | C110 | ≤0.35 | ≤0.005 | ≤1.2 | ≤0.020 | ≤0.99 | 0.4–1.5 | - | 0.25–1.00 | - |
| | P110 | - | ≤0.030 | - | ≤0.030 | - | - | - | - | - |

### 3.2. Metallographic Analysis

According to ASTM E45-18a, GB/T10561-2005 [28], and ASTM E112-13 [29] standard, the samples are machined into 10 mm × 15 mm × thickness to analyze metallographic structure and the non-metallic inclusion and the microstructure, as shown in Figure 2. The microstructure of the failed tubing is tempered sorbite (Figure 2a). The diameter of the inclusion is 12 μm and it becomes an $Al_2O_3$ inclusion mainly composed of Al and O (Figure 2b).

### 3.3. Hardness Analysis

Referencing standard ISO 11960-2011, the annular specimens with a thickness of 10 mm are taken from the thinned tubing, and then the hardness of the samples (Figure 3) is tested by using a Rockwell hardness tester. The allowable error of calibration tool should be in the range of approximately 2%, and the standard deviation was given from three measurements [30]. The hardness of the thinned tubing is between 25.5 HRC and 30.1 HRC, and the average hardness is 27.89 HRC (Figure 4). According to the standard NACE MR0175, the hardness of P110S steel in $H_2S$-containing surroundings should be lower than 37 HRC [31]. Therefore, the hardness of tubing does not exceed the standard requirements, which means that the hardness of oil pipe is qualified.

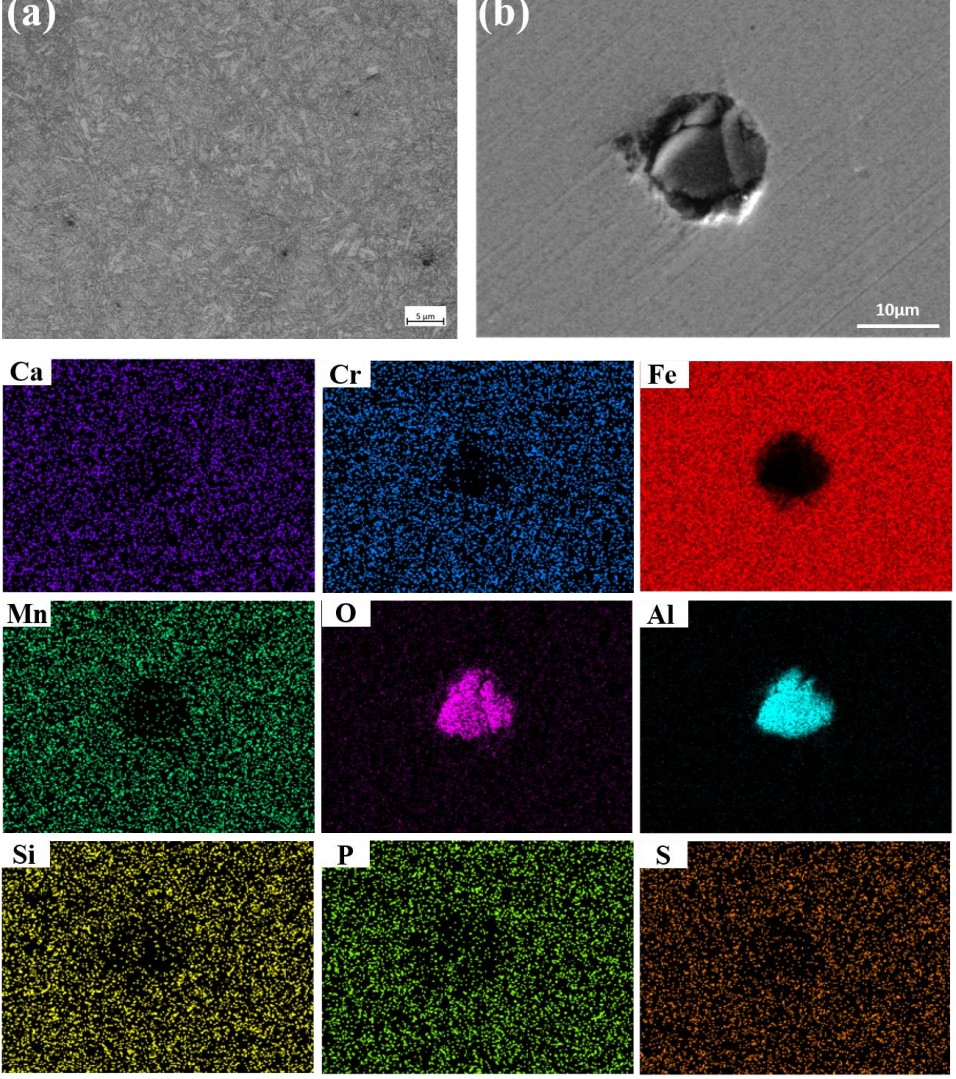

**Figure 2.** Metallographic structure and non-metallic inclusion of failed tubing. (**a**): Metallographic structure; (**b**): non-metallic inclusion.

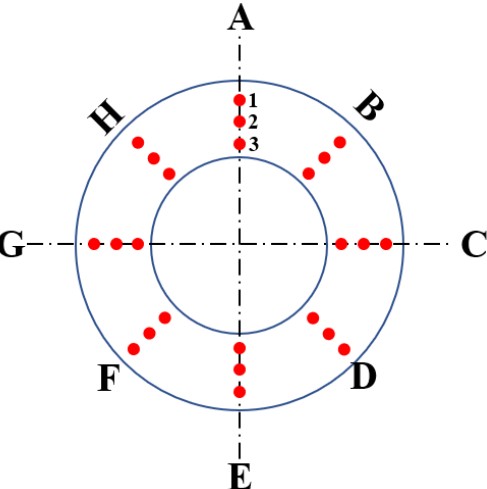

**Figure 3.** Hardness measurement point in A–H direction.

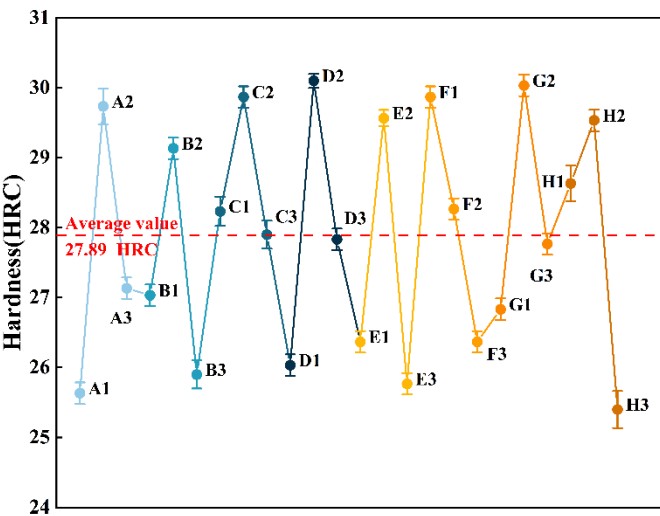

**Figure 4.** Hardness test results at A1–H3 points.

*3.4. Tensile Test*

To determine the tensile strength of the thinned tubing, the tensile property test was carried out according to the standard ISO 6892-1:2009 [32]. The lath-like tensile samples were cut from the thinned tubing, and the tensile test was performed at 25 °C with an MTS810 tensile tester at a stretching speed of 5 mm/min, as shown in Table 2. According to API Spec 5CT-2011 standard, the fracture elongation, yield, and tensile strength of thinned tubing satisfies standards [33].

**Table 2.** Testing results of tubing body tensile properties.

| Specimen No. | Tensile Strength (MPa) | | Yield Strength (MPa) | | Yield Ratio | Fracture Elongation (%) | |
|---|---|---|---|---|---|---|---|
| | Measured Value | Average Value | Measured Value | Average Value | | Measured Value | Average Value |
| I | 881 | | 830 | | 0.95 | 10.6 | |
| II | 875 | 878 | 836 | 836 | 0.95 | 10.1 | 10.3 |
| III | 878 | | 842 | | 0.95 | 10.2 | |
| API-5CT | | ≥862 | | 778–965 | - | | - |

*3.5. Morphology and Composition Analysis of Corrosion Products*

Based on the corrosion properties of the thinned tubing, the parts of tubing below the packer are selected for observing morphology and analysis composition, as shown in Figure 5.

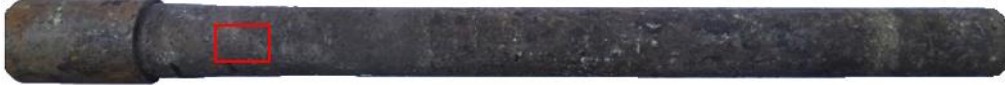

**Figure 5.** Sampling area of failed tubing.

3.5.1. Surface Morphology Observation of Corrosion Products

It is observed that the outer wall of the tubing is uniformly corroded and severely scaled, and a layer of black gray scaling products is attached to the outer wall. The adhesive tape is used to stick down corrosion products to observe the morphology due to the fact that the outer films of corrosion product on the outer wall easily fall off, as shown in Figure 6. The scraped corrosion products are extremely loose and many cracks are also observed, which is a channel for corrosion solution.

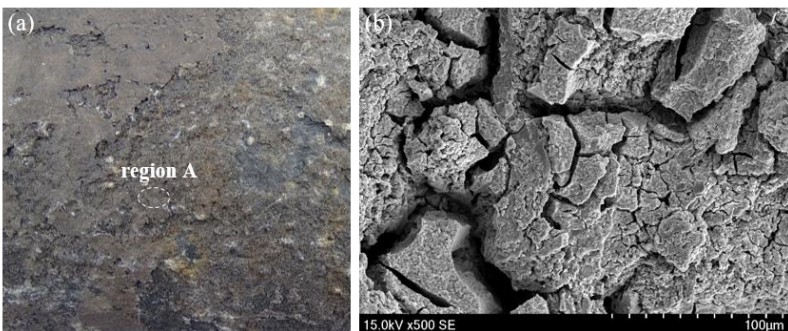

**Figure 6.** Sticking position of corrosion scales on the outer wall (**a**) and SEM images of corrosion stuck from the inner wall of tubing (**b**).

The SEM images, as shown in Figure 7, demonstrate that the corrosion product films on the inner wall of tubing show a layered structure. The corrosion product layer is thick, and there are a lot of micro-cracks on the surface of the corrosion product films. $CO_2$ corrosion trace is found at high magnification, a small amount of $FeCO_3$ grains have obvious edges and corners, but most of the grains have edges and corners dissolved. In addition, a large number of irregular spherical particles can be seen attached to the corrosion product film.

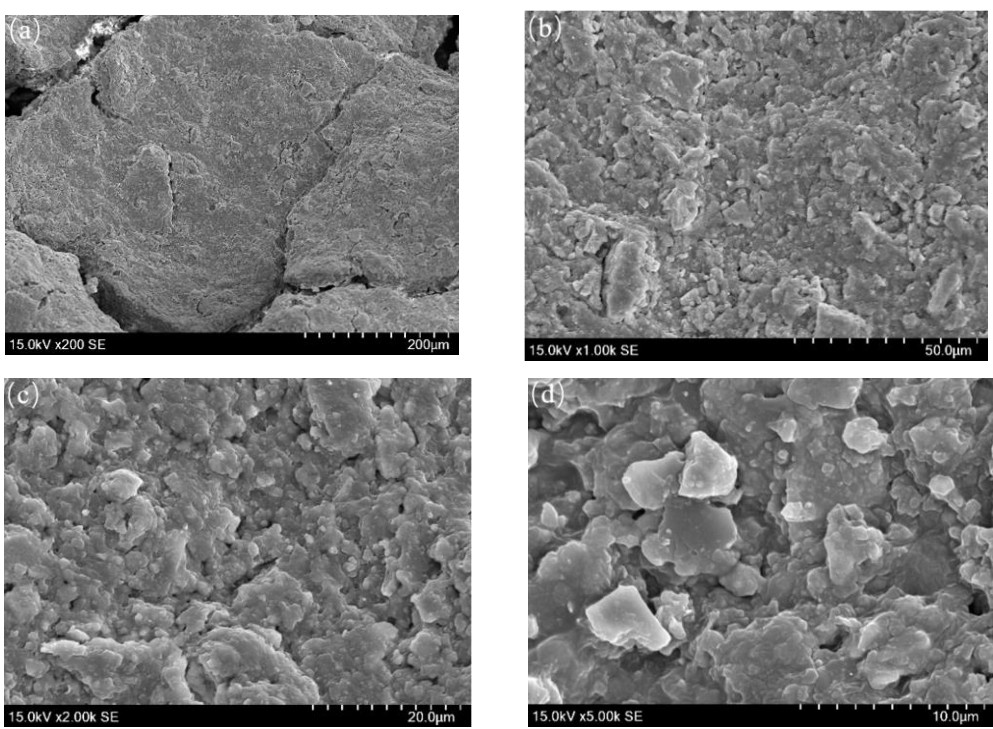

**Figure 7.** SEM images of surface morphology on inner wall of tubing. (**a**): 200 × SEM; (**b**): 1000 × SEM; (**c**): 2000 × SEM; (**d**): 5000 × SEM.

The SEM images in Figure 8 show that the corrosion scales on the outer wall of the tubing include two kinds of morphology (Figure 8a). The first one has some holes on the scales (Figure 8b–d). The second kind of the corrosion scales, which are very dense, are formed by the typical cubic crystal (Figure 8c). The corrosion morphology of the inner and outer wall of tubing is quite different, while the composition of those are similar according to XRD results. Due to the erosion of the corrosion scales on the inner wall by the produced liquid, the corrosion scales become flaky. However, the outer wall of tubing has less fluidity than the inner wall before the packer leakage, and the $FeCO_3$ crystal formed on the outer wall is relatively intact.

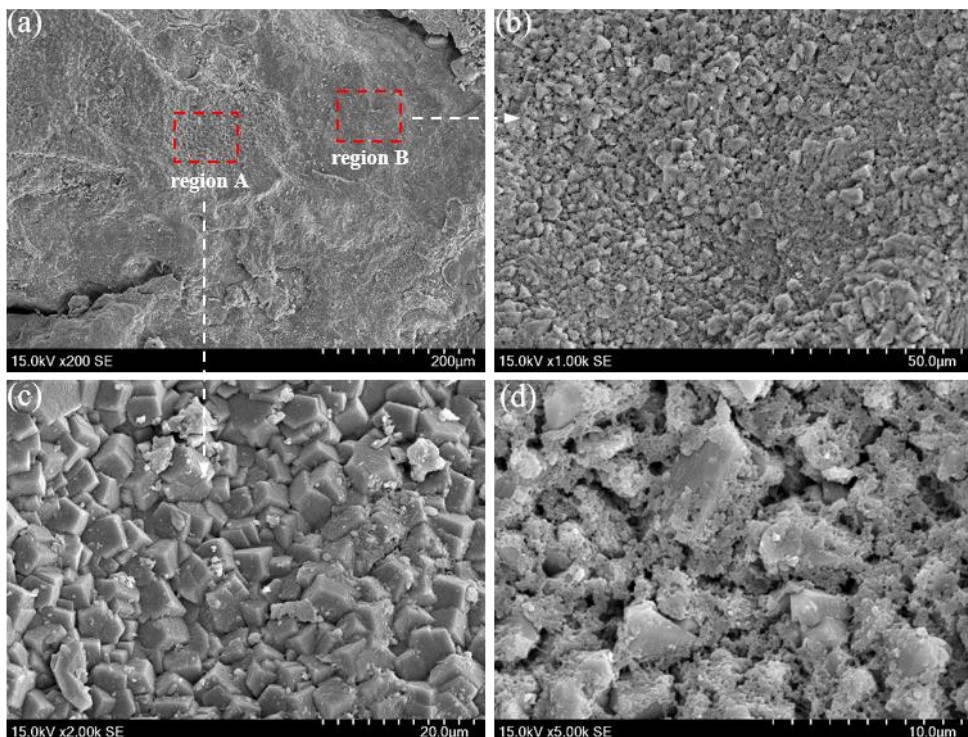

**Figure 8.** SEM images of surface morphology on outer wall of tubing. (**a**): 200 × SEM; (**b**): 1000 × SEM; (**c**): 2000 × SEM; (**d**): 5000 × SEM.

As it shows in Figure 9, the component of the corrosion scales on the inner wall of the tubing is more complex, while that on the outer wall of the tubing is less so. Moreover, the salt content in the corrosion scales that stuck from the inner wall is less, while the salt content is rich in inner wall, indicating that the salts are concentrated in the inner layer. The corrosion scales on outer wall consist of Ca, Fe, C, Cl, and O, implying that the $CO_2$ corrosion and calcium carbonate scales may be generated on the inner wall of the tubing [34,35]. It is worth noting that S elements are not observed in EDS results. Moreover, traces of Ba and Si elements are observed according to EDS results, which come from the drilling fluid.

### 3.5.2. Cross-Section Morphology Observation of Corrosion Products

The cross-section morphology and corresponding element distribution images of the inner wall are displayed in Figure 10. The corrosion product films on the inner film are extremely loose and porous. In addition, a corrosion pit with the depth of 19.03 μm is also observed in Figure 10b. According to the element distribution of the corrosion scales, the main elements are O, C, and Fe. A trace of Ba, S, and Sr elements are distributed in the scales (Figure 10c–h). It is remarkable that the C element accumulates in the upper layer, while the O element accumulates in the lower layer (Figure 10c,f).

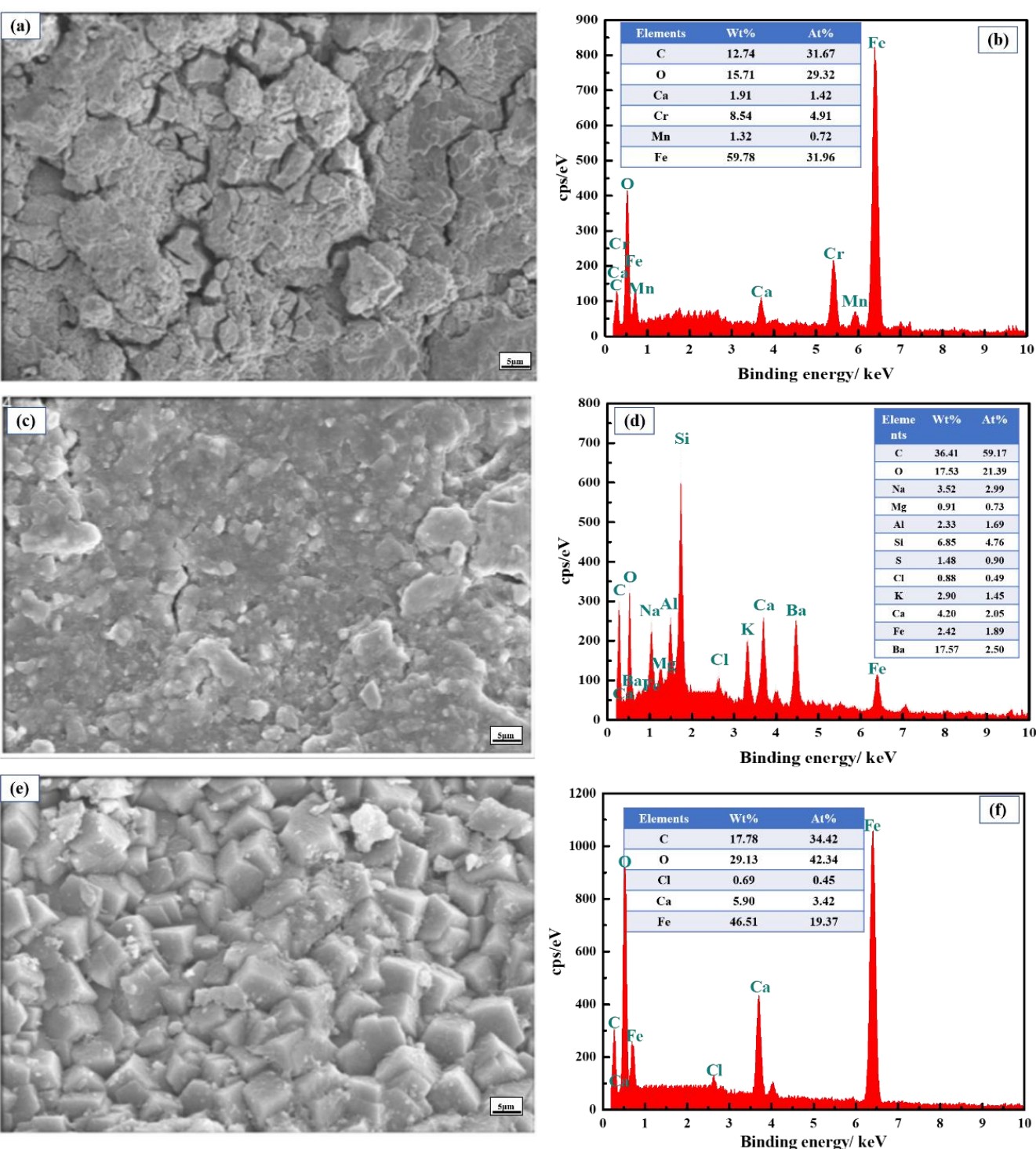

**Figure 9.** EDS images of corrosion scales on failed tubing. (**a**,**b**): Stuck corrosion scales; (**c**,**d**): inner wall; (**e**,**f**): outer wall.

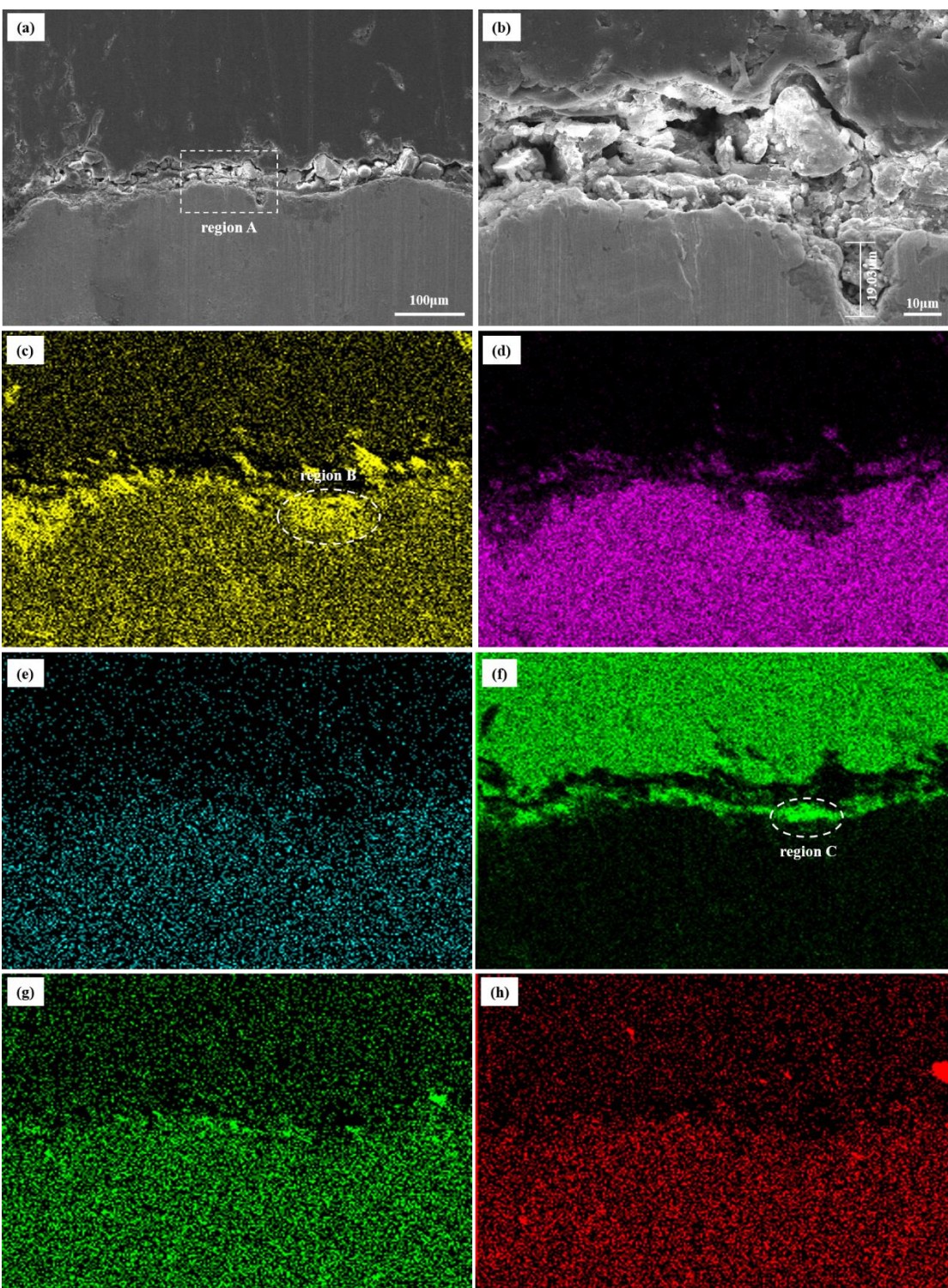

**Figure 10.** SEM/EDS images of corrosion scales on inner wall. (**a**,**b**): Cross-section SEM surface morphologies; (**c**): oxygen; (**d**): ferrum; (**e**): barium; (**f**): carbon; (**g**): sulfur; (**h**): strontium.

### 3.5.3. Composition Analysis of Corrosion Products

X-ray diffraction (X′ Pert MPD PRO) was used to analysis the attachments taken from the surface of tubing wall (Figure 11). The main corrosion products of the inner and outer wall are $FeCO_3$ and $BaSO_4$, while $CaCO_3$ still exists in the corrosion products of the outer wall. In addition, no sulfide is found in the XRD results, and the results are consistent with

the EDS results. Hence, it can be judged that the corrosion process of the failed tubing is $CO_2$ corrosion.

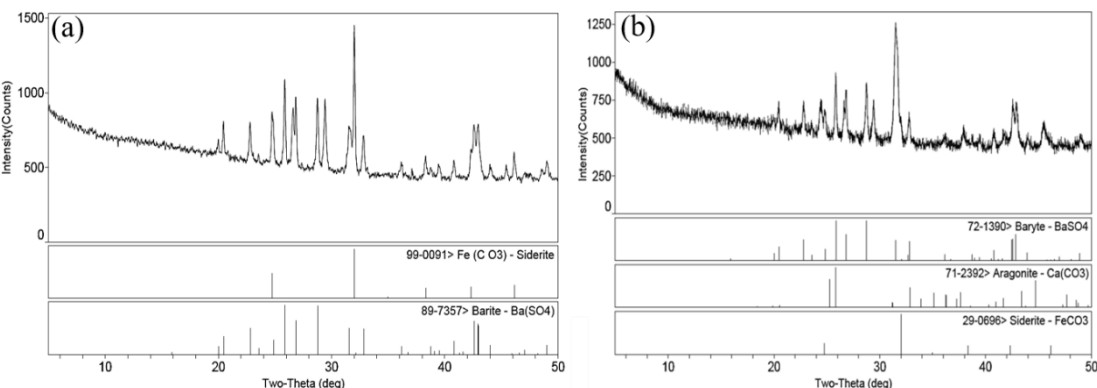

**Figure 11.** XRD results of adhesives on the inner (**a**) and outer (**b**) wall of tubing.

### 3.5.4. Pitting Analysis of Inner Wall

Taking samples on the inner wall of the tubing (Figure 12a), the corrosion products on the surface of the sample were cleaned with the de-filming solution. Then, a 3D confocal laser microscope (Keyence VK-X250K, USA) was used to analyze the pitting on the inner wall of the failed tubing. Figure 12b shows the 3D morphology of the inner wall of tubing after removing corrosion products. Lots of pitting is observed on the surface, and the pitting depth reaches 35 μm.

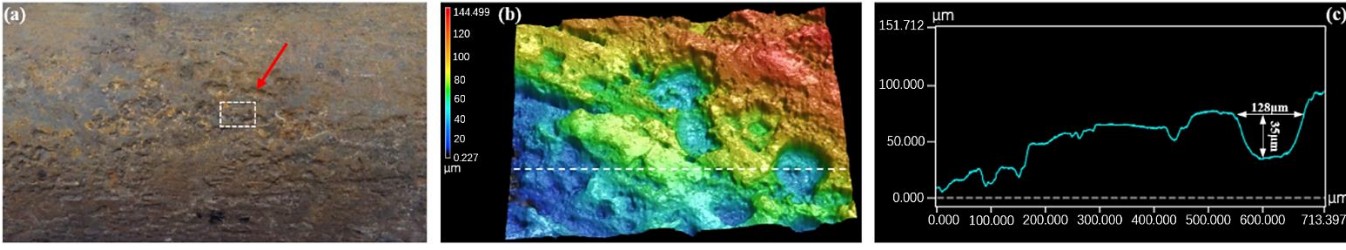

**Figure 12.** Three-dimensional morphology of the inner wall of tubing after removing corrosion products. (**a**): Sampling location; (**b**): 3D morphology; (**c**): pitting depth.

### 3.6. Prediction of Corrosion Life of Tubing

According to Equations (1)–(6), the remaining safety life and safety factor against the internal pressure of the tubing are predicted. Figure 13 describes the safety factor of anti-inner pressure and service life on tubing. As it shows, the safety factor of anti-inner pressure on tubing below the packer rapid decreases when the service life increases, while that of the tubing above the packer drops slowly (Figure 13a). The safe service life of the tubing above the packer is 20.6 years, while that of tubing below the packer is only 4.6 years (Figure 13b). Hence, a corrosion-resistant alloy is required for tubing below the packer.

According to the survey data of a well in the same block, the well was completed in February 2015, and the tubing was found to be corroded and fractured in early 2019 (Figure 14a,b). The actual service life of the tubing is about 4 years. The tubing life prediction model has good reliability.

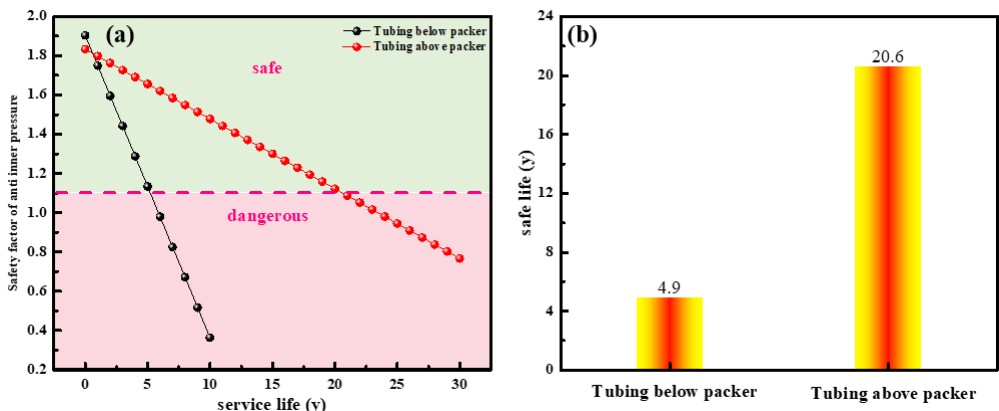

**Figure 13.** Safety factor of anti-inner pressure (**a**) and service life (**b**) of tubing.

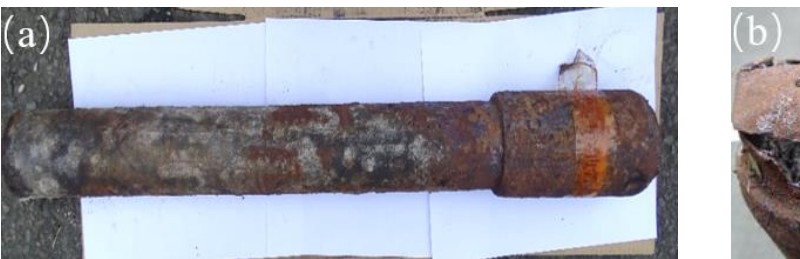
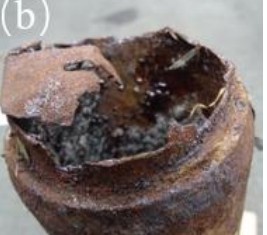

**Figure 14.** Fracture of tubing due to corrosion. (**a**): Tubing body; (**b**): local fracture.

## 4. Discussion

The content of P and S elements in P110S steel tubing of a failed well was lower than the standard minimum value. The metallographic structure of tubing was tempered sorbite. The uniform structure, fine grains, and the size of non-metallic inclusions were normal. It shows that the physicochemical properties of the tubing met the regulations of relevant standards. In addition, the average hardness of the tubing was 27.89 HRC, which was lower than the upper limit value of 37 HRC of P110S steel produced by domestic and foreign iron and steel enterprises. The results show that the quality of the tubing was qualified and had no special impact on the corrosion.

The failed tubing was applied in a complicated acidic corrosive environment, in which $H_2S/CO_2$ coexist, based on wellbore data. However, no sulfur-related corrosion products are detected in XRD results. Ikeda et al. demonstrated the influence of temperature on steel corrosion in a $CO_2$–$H_2S$ environment, and they discovered that $CO_2$ controlled the whole corrosion process when the ambient temperature exceeded 150 °C [36]. Moreover, $Ca^{2+}$ and $Ba^{2+}$ ions in the produced liquid increased the hardness of the produced liquid, resulting in the decrease in the amount of $CO_2$ and $H_2S$ dissolved in the aqueous solution. The service temperature of the failed tubing is 160 °C, and a lot of $Ca^{2+}$ and $Ba^{2+}$ ions exist in the produced fluid in the failed tubing. Hence, according to the test data and the actual service environment of the tubing, we judged that the corrosion of the failed tubing was controlled by $CO_2$ [37].

$BaSO_4$ and $CaCO_3$ are the common sulfate scale and carbonate scale in the oilfield (Equations (6) and (7)) [38,39]. The calcium carbonate precipitates and adsorbs on the outer wall of tubing. The most probable reason is that the annulus protection fluid in the annulus is made of the produced water, and traces of $Ca^{2+}$ ions exist in the produced water. $CO_2$ gas leaks into the annulus, generating $CaCO_3$ precipitation when the packers leak. Barium sulfate is the main component of drilling fluid. Barium sulfate is attached to the inner wall, which may be due to more drilling fluid leaking into the formation during drilling and returning from the produced well during the process of gas production.

The free $CO_2$ in formation water is in balance with that in $CO_3^{2-}$, $HCO_3^{-}$ ions. The pressure of the wellbore drops, which makes the free $CO_2$ in the solution overflow when

the formation water enters the wellbore from the formation. In order to maintain the equilibrium, $Ca(HCO_3)_2$ dissolved in water decomposes and produces $CO_2$ to supplement free $CO_2$ in solution [40]. However, the decomposition of $Ca(HCO_3)_2$ also produces $CaCO_3$ at the same time. Moreover, the decrease in temperature results in the decrease in solubility of $CaCO_3$ and $BaSO_4$, leading $CaCO_3$ and $BaSO_4$ to continuously accumulate and adsorb to the inner wall.

$$Ca^{2+} + CO_3^{2-} \rightarrow CaCO_3 \tag{6}$$

$$Ca^{2+} + 2HCO_3^- \rightarrow CaCO_3 + CO_2 + H_2O \tag{7}$$

The under-deposit corrosion mechanism of the failed tubing is described in Figure 15. The produced liquid washes away the loose corrosion scales including $FeCO_3$ and $CaCO_3$, and $BaSO_4$ in some areas. Some corrosion scales are washed away, exposing the matrix, which is not corroded by the corrosive environment, becoming a fresh corrosion anode to further corrode (Figure 15a). For one thing, the damage region and undamaged region form the large cathode (the undamaged area)–small anodic corrosion (the damaged area), which promotes the advancement of corrosion pits. As the corrosion pit deepen, the mass transfer resistance of corrosive ions increases with the new formed $FeCO_3$, and the corrosion pit is gradually blocked by $CaCO_3$ products, thus, composing a concentration cell (Figure 15b). In addition, chloride ions with a small radius easily penetrate the newly formed $FeCO_3$ products and migrate to the pit (Figure 15c). As a catalyst, chloride ions accelerate the dissolution of $Fe^{2+}$, resulting in a deep corrosion pit. Also of note is the fact that the falling off scales makes a large area of the matrix become a fresh corrosion surface, hindering the longitudinal development of pitting corrosion, and promoting the overall corrosion of the tubing.

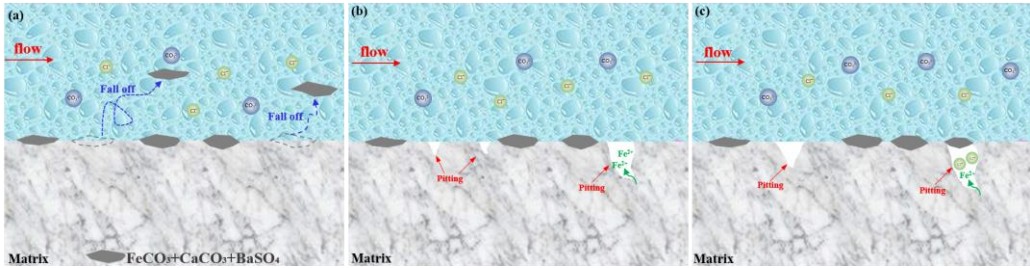

**Figure 15.** Schematic diagram of under-deposit corrosion mechanism in failed tubing. (**a**): Loose corrosion scales fall off; (**b**): concentration cell; (**c**): chloride ions penetrate $FeCO_3$.

## 5. Conclusions

(1) The results of the chemical composition test, metallographic structure test, and hardness test of the failed tubing meet a criterion of ISO 11960-2011. The microscopic structure of P110S steel is sorbic-acid-tempered, and the non-metallic inclusion is ASTM D0.5. The quality of tubing is qualified;

(2) Due to the ponding at the bottom hole in the later stage, the tubing under the packer is seriously corroded and thinned. The tubing below the packer suffers from water-accumulation corrosion, while the tubing above the packer suffers from water-carrying corrosion. The wellbore with water should be drained in time to avoid water-accumulation corrosion;

(3) $CO_2$ corrosion is the main corrosion of failed tubing. The failed tubing is seriously damaged by under-deposit corrosion in the local areas. The raised corrosion scales and scales on the surface of steel are washed away by the produced liquid to form pitting, and $FeCO_3$ and $CaCO_3$ accumulate on the top of pitting to form corrosion under the scale;

(4) A novel model that was specific to the service life of tubing in terms of the wall thickness of tubing was established. The calculated results indicate that the safe service life of tubing above the packer is 20.6 years, while that of tubing below the

packer is 4.9 years. A corrosion-resistant alloy is required for tubing below the packer. It provides guidance for the safe servicing of tubing.

**Author Contributions:** Conceptualization, W.L. (Wu Long) and X.W.; methodology, X.W. and X.C.; software, W.L. (Wei Lu) and S.C.; validation, X.W., H.H. and W.L. (Wei Lu); formal analysis, X.W. and L.L.; investigation, M.Z.; resources, M.Z. and S.C.; data curation, W.L. (Wei Lu) and X.W.; writing—original draft preparation, W.L. (Wu Long); writing—review and editing, X.W.; visualization, H.H. and W.L. (Wei Lu); supervision, W.L. (Wu Long) and X.C.; All authors have read and agreed to the published version of the manuscript.

**Funding:** This research received no external funding.

**Data Availability Statement:** Not applicable.

**Conflicts of Interest:** The authors declare no conflict of interest.

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
