# Peer review of "Primary Corrosion Cause of P110S Steel Tubing Corrosion Thinning in CO2–H2S Well and Its Remaining Life Prediction"

_processes, doi:10.3390/pr11020333_

Round 1

Reviewer 1 Report

Taking an oilfield gas well as an example, the authors analyze the reasons for the thinning of gas well tubing in CO2-H2S environment by combining the tests of chemical composition, metallographic, hardness, mechanical properties, and the corrosion layers observation. Then, according to these analyses, the cause and mechanism of tube failure are discussed, and finally the safe service life model based on thickness measurement data is used to predict the remaining life of tubing.

This paper studies the durability problem in actual engineering where chemical corrosion exists, and has reference value for the corresponding project. However, the manuscript has obvious deficiencies in the research arguments and the discussion of the new methodology, and the authors must carefully revise before publication:

1.        Tubing size data in Figure 1 do not match, for example: 273.1mm/2-8.89mm is not equal to 193.7mm/2, the authors should provide an explanation. And the authors should also provide a description of the original size of the gas well tubing.

2.        The mechanism of tube failure and the conditions for determining life is not explained in the text. Why the yield described in Equation (3) is used as a condition for the failure of the tubing, the authors should give a reasonable explanation as to this equation can effectively judge tubing failure.

3.        The text does not explain the stress state of the tubing shown in Figure 1, and the Equations (1) ~ (3) only mentions the circumferential stress of the tubing, which the author should explain. The authors mention the relevant criteria in the article, and since it is not necessarily possible for the reader to consult with all the standards listed in the text. It is recommended that the authors provide a concise explanation of the main basis for the standards.

4.        The article suggests that "after the tubing is used for t time, the wall thickness of the tubing is δ (δ0=δ-vt)", and mentions " the reduction of wall thickness of tubing above the packer is only 0.125mm/y, while that of tubing below the packer is as high as 0.445mm/y”. Are these data given according to historical records? Or is based on the final measurement conjecture?  

5.        In the paper, the authors saidAccording to the load of the tubing, the relationship between the residual internal pressure safety factor and the service time was obtained". Is this relationship considering of the influence of microcracks? When there are a large number of microcracks on the surface of the tubing (Figures 7 and 8 and related explanations). Here "the relationship between the residual internal pressure safety factor and the service time" is it still correct?

6.        The paper mentions "A novel model" or "A calculation model", but the details about where innovations and theoretical basis of the model are not clearly discussed or explained.

7.        It is recommended that the authors supplement the reasonableness demonstration or validation of the life prediction of gas well tubing.

Author Response

The reviewer's comments have been replied. Please refer to the attachment for details.

Reviewer 2 Report

This article has a good scientific value in relation to the corrosion of tubing. But before the final decision, it needs many changes and corrections. After correcting the manuscript based on the comments, I will announce my opinion regarding the acceptance or rejection of the article. Consider all the comments below and highlight the changes.

1- The grammar and language structure of the article is inappropriate in some parts. It should be revised.

2- The type of tubing used must be stated in the title.

3- Do not use long sentences in the abstract. The investigated parameters should be briefly stated in the abstract. In the abstract, state what tests were used and what results were obtained. Extra sentences should be deleted. 

4- The introduction is incomplete. In the introduction, the relationship between metallurgical parameters, microstructural changes and corrosion behavior has not been investigated. Introduction is not acceptable in its current format. Use the following articles to complete this section:

- https://doi.org/10.1016/j.ijpvp.2022.104759 , - https://doi.org/10.1016/j.ijhydene.2015.04.114 , - https://doi.org/10.1016/j.jmrt.2021.07.118 , - https://doi.org/10.1016/j.jmst.2013.10.018 , - https://doi.org/10.1016/j.ijpvp.2019.02.019  

5- All formulas used must be specified from which reference they are derived.

6- State the standard used for corrosion tests.

7- The results of mechanical and corrosion tests should be reported along with the standard deviation.

8- Corrosion morphology should be thoroughly analyzed.

Author Response

The reviewer's comments have been replied. Please refer to the attachment for details

Reviewer 3 Report

Reviewer Recommendation and Comments for manuscript processes-2104257-peer-review-v1.pdf with the title: “Primary corrosion cause of P110S tubing corrosion thinning in 2 CO2-H2S well and its remaining life prediction”, with follow authors Wu Long, Xi Wang, Huan Hu , Wei Lu , Lian Liu , Miaopeng Zhou , Sirui Cao, Xiaowen Chen.

 This manuscript predicted the safe service life of tubing based on the collected thickness measurement data. The analyses of tubing samples consist chemical composition, metallographic structure and hardness. The failed tubing was analyzed by using material property testing, SEM, EDS and XRD.

This is a relevant study dealing with the problem of cause of tubing thinning of gas wells in Northwest Oilfield. The authors state that: “A novel model which was specific to the service life of tubing in terms of the wall thickness of tubing was established”. This research is interesting due to predict safe service life of tubing.

The text is clearly written. The structure, content, and concept of the research work as well as the achievements, correspond to the new unpublished study but some improvement are necessary. The English is fine and the paper is clearly written.

The abstract is clear and in the proper way present the scope, experiment, achievements, but future research directions are not mentioned.

The introduction and literature review consists of relevant literature data (18 articles) and properly analyzed previous studies in the fields of research.

Tables, Figures, and math equations are properly presented but some improvement are mandatory. On some figures missing explanation

The conclusion is short, but clearly explains the achievements of this original scientific research but some improvement can be done.

The main comments that I find useful for improving the quality of the article are presented below:

Finding 1: (Line 32 and 33). In all manuscript authors do not properly write “et al”, it should be “et al.”

Finding 2: Figure 1 is not mentioned in text and it is not explained in line 92 and nomenclature of all data on figure are missing. Explanation is mandatory.

Finding 3: Lines 120-131. Formulas are not properly formatted.

Finding 4: Line 171. It is not Tab.5 it is Tab.2. All other numeration of Figures should be corrected.

Finding 5: Line 186. Explain why “Region A” is important of figure.

Finding 6: Line 192. They are missing numeration of figures and explanation as on line 154.

Finding 7: Line 204. Explain why region A and B are important on figure and explain in text.

Finding 8: Line 2284. Explain why region A and C are important on figure and explain in text.

Finding 9: Lines 245 and 247. It should be 13.b.

Finding 10: Lines 252-257. It is not clearly explained new model and results. What is the new model and achievements?

Finding 11: Line 137 and line 261 consist word “discussion”. It is not acceptable.

Finding 12: Line 272 it is missing a reference for author Ikeda et al.?

Finding 13: Formulas 4 and 5 are not mentioned in text of manuscript. Could you connected it?

Finding 14: Line 3162. They are missing explanation of figures as on line 154.

Finding 15: In conclusion, number (4) it is not clear proposed model. Line 17-18, paragraph 3.6 and state (4) from conclusion do not clearly present new model and achievements. Authors should improve this achievements.

 I suggest major revision of this manuscript before publishing.

Author Response

(The authors gave the same response as above.)

Round 2

Reviewer 1 Report

The authors have revised the text and improved the discussion in the text, and answered the questions of the reviewer.

The paper performed corrosion-related strength analysis of the reduced service life of tubing due to corrosion thinning, which is more engineering than scientifically significant. In this paper, the corrosion rate of the pipeline is set as a constant value, the thickness of the corrosion layer is uniform, the microcracks generated by the corrosion process are omitted, and the material yield is used as the criterion for tubing failure, all of these no convincing evidence is given in discussions. The relationship between pipeline pressure and depth, whether there are dynamic loads or cyclic loads, is also lacking. There are still errors in the units of some physical quantity in the text after modification. The reviewer recommends that authors carefully revise and supplement the paper before publication.

Author Response

We have replied to the experts' opinions one by one. Please refer to the attachment for details.

Reviewer 2 Report

Although the authors have made changes to the article, the quality of this manuscript is very low. Also, the authors did not respond properly to my comments. I will send my comments to the authors again. After correcting the manuscript based on the comments, I will announce my opinion regarding the acceptance or rejection of the article. Changes must be highlighted:

1- The grammar and language structure of the article is inappropriate in some parts. It should be revised.

2- The type of tubing used must be stated in the title.

3- Do not use long sentences in the abstract. The investigated parameters should be briefly stated in the abstract. In the abstract, state what tests were used and what results were obtained. Extra sentences should be deleted.

4- The introduction is incomplete. In the introduction, the relationship between metallurgical parameters, microstructural changes and corrosion behavior has not been investigated. Introduction is not acceptable in its current format. Use the following articles to complete this section:

https://doi.org/10.1016/j.ijpvp.2022.104759

https://doi.org/10.1016/j.ijhydene.2015.04.114

https://doi.org/10.1016/j.jmrt.2021.07.118

https://doi.org/10.1016/j.jmst.2013.10.018

https://doi.org/10.1016/j.ijpvp.2019.02.019

5- All formulas used must be specified from which reference they are derived.

6- State the standard used for corrosion tests.

7- The results of mechanical and corrosion tests should be reported along with the standard deviation.

8- Corrosion morphology should be thoroughly analyzed. 

Author Response

(The authors gave the same response as above.)

Reviewer 3 Report

Your corrections are fine and accepted.

I suggest to publish this manuscript.

Author Response

Thank you for your approval.

Round 3

Reviewer 2 Report

I recommend "Publish As Is".